# Generate What You Prefer: Reshaping Sequential Recommendation via Guided Diffusion

**Zhengyi Yang**[‡] **Jiancan Wu**[‡*] **Zhicai Wang**[‡] **Yancheng Yuan**[§*] **Xiang Wang**[‡†] **Xiangnan He**[‡†]

[‡]University of Science and Technology of China
[§]The Hong Kong Polytechnic University
{yangzhy,wangzhic}@mail.ustc.edu.cn
{wujcan,xiangwang1223,xiangnanhe}@gmail.com
yancheng.yuan@polyu.edu.hk

## Abstract

Sequential recommendation aims to recommend the next item that matches a user's interest, based on the sequence of items he/she interacted with before. Scrutinizing previous studies, we can summarize a common **learning-to-classify** paradigm — given a positive item, a recommender model performs negative sampling to add negative items and learns to classify whether the user prefers them or not, based on his/her historical interaction sequence. Although effective, we reveal two inherent limitations: (1) it may differ from human behavior in that a user could imagine an oracle item in mind and select potential items matching the oracle; and (2) the classification is limited in the candidate pool with noisy or easy supervision from negative samples, which dilutes the preference signals towards the oracle item. Yet, generating the oracle item from the historical interaction sequence is mostly unexplored. To bridge the gap, we reshape sequential recommendation as a **learning-to-generate** paradigm, which is achieved via a guided diffusion model, termed **DreamRec**. Specifically, for a sequence of historical items, it applies a Transformer encoder to create guidance representations. Noising target items explores the underlying distribution of item space; then, with the guidance of historical interactions, the denoising process generates an oracle item to recover the positive item, so as to cast off negative sampling and depict the true preference of the user directly. We evaluate the effectiveness of DreamRec through extensive experiments and comparisons with existing methods. Codes and data are open-sourced at https://github.com/YangZhengyi98/DreamRec.

## 1 Introduction

Sequential recommendation has long been a fundamental and important topic in many online platforms, such as e-commerce, streaming media, and social networking [1–3]. Its core task is to recommend the next item that matches user preference, based on the sequence of items he/she interacted with before. Scrutinizing recent research on sequential recommendation [4–9], we may discern a common **learning-to-classify** paradigm: given a sequence of historical items and a target (truly positive) item, a recommender model first performs negative sampling to append the historical interactions with some non-interacted (possibly negative) items, and then learns to classify the positive instance from the sampled negatives. Along with this line, extensive studies are conducted on evolving the model architecture (*e.g.,* recurrent neural networks [4], convolutional neural networks

---

[*]Jiancan Wu and Yancheng Yuan are corresponding authors.

[†]Xiang Wang and Xiangnan He are also affiliated with Institute of Artificial Intelligence, Institute of Dataspace, Hefei Comprehensive National Science Center.

37th Conference on Neural Information Processing Systems (NeurIPS 2023).

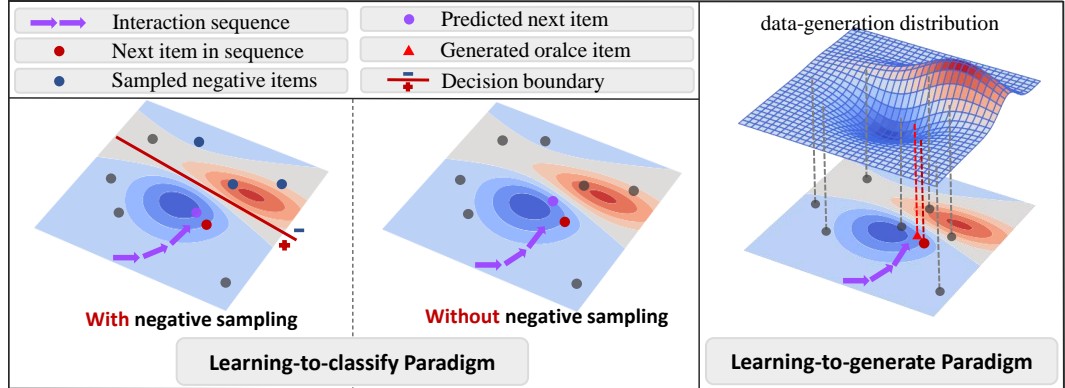

Figure 1: Illustration of learning-to-classify and learning-to-generate paradigms of sequential recommenders *w.r.t.* the treatment of item embeddings. In the learning-to-classify paradigm, the negative sampling technique is typically employed to classify the positive instance from the sampled negatives (*left* subfigure); Without negative sampling, all items are naively viewed as positive, undermining recommendation performance (*middle* subfigure). In contrast, the learning-to-generate paradigm leverages observed interactions to capture the underlying data-generation distribution, enabling it to generate the oracle item beyond the candidate set (*right* subfigure).

[10, 11] and Transformers [5, 6]) and auxiliary tasks (*e.g.,* causal inference [7, 12], contrastive learning [8, 13], robust optimization [9]), to enhance the classification ability and characterize the user preference better.

Clearly, this learning-to-classify paradigm seeks to predict whether an item in the candidate set (*i.e.,* positive and sampled negative items) aligns with the user preference evidenced by the historical item sequence. Despite its success, we reveal two limitations inherent in this paradigm:

- It may simplify human behavior — a user could imagine an **oracle item** in mind, and then compromise to a real item that best matches the oracle. By "imagine", we mean that the oracle item is the one that a user would ideally like to interact with next, softly absorbing and representing his/her preference from previous interactions. By "compromise", we mean that the oracle item, conceivably, is not a concrete and discrete instance limited in the candidate set, while the real item interacted with next exemplifies the oracle. Considering the example in Figure 1, the red triangle ▲ is the oracle item ingratiating with the interaction sequence, while the red circle ● is the real item close to the oracle item and satisfying the user.

- Classification upon the candidate set is a circuitous way to characterize the user preference, to address the one-class problem posed by the availability of only positive samples [14]. Yet, as the selected negatives are confined in a small candidate set (*e.g.,* the blue circles ● in Figure 1), their contrasts with the positive sample (*e.g.,* the red circle ● in Figure 1) coarsen the decision boundary between what a user likes and dislikes, leaving the item space mostly unexplored. Worse still, simple negative samples may be far away from the positive item, thus failing to contribute sufficient supervision signals to the model learning [15]; whereas, the overly complex ones might be falsely negative and inject noises into the model [14, 16, 17]. Therefore, such classification signals are hard to control to depict the oracle item.

To resolve these limitations, we reshape sequential recommendation from the perspective of **learning-to-generate** instead. The basic idea is to describe the underlying data-generation distribution based on the historical item sequence, directly generate the oracle item that softly represents the user preference, and infer the real items most matching the oracle, as shown in Figure 1 (*right* subfigure). Conceptually, generating the oracle item enables us to go beyond the scope of all concrete and discrete items, and encouraging its agreement with the positive items allows us to get rid of negative sampling. Towards this end, we are setting our sights on the diffusion generative models [18–22]. The key idea of diffusion generative models is to gradually convert the data into noise, and generate samples by the parameterized denoising process. Yet, generating oracle items in sequential recommendation via diffusion models is mostly unexplored.

To bridge this gap, we propose a simple yet effective approach **DreamRec**, which uses the guided diffusion idea to directly generate the oracle item tailor-made for the user history. Specifically, given a sequence of historical items, the forward process of DreamRec progressively adds noise to the target item's representation and nearly leads to a complete noise, while the reverse process gradually denoises a Gaussian noise guided by history to generate the oracle item aligned with the historical interactions. Wherein, we establish the history guidance by applying a Transformer encoder on the historical item sequence, so as to make the oracle item specialize for each sequence. As a result, DreamRec enjoys the merit of directly modeling the user preference, without relying on classification and negative sampling. On three benchmark datasets, we evaluate the effectiveness of DreamRec through extensive experiments and comparisons with existing methods [4, 10, 5, 7, 12, 8].

## 2   Related Work

**Sequential recommendation**, which captures user perference and recommends the next items based on the historical interactions, has garnered increasing attention in academia and industry [5, 4, 23]. Most existing studies approach the task under a learning-to-classify paradigm, where the decision boundary separates the positive instance from the sampled negatives. To enhance the classification ability of sequential recommenders, existing efforts can be broadly categorized into two branches. The first branch centers around leveraging complex model architectures, such as recurrent neural networks [4], convolutional neural networks [10, 11], Transformer encoders [5, 6, 24], to encode the interaction sequences and better characterize user preferences. While the second research line is centered around the inclusion of diverse auxiliary learning tasks, including causal inference [7], contrastive learning [8, 13], and robust optimization [9].

**Diffusion Models** have emerged as a prevailing approach for various generative tasks, including image synthesis [25, 18], text generation [22], and molecule design [26]. Their popularity stems from the ability to precisely approximate underlying data generation distribution and offer more stable training compared to other generative models like GANs [27, 28] and VAEs [29, 30]. Three primary formulations exist in the literature [31]: denoising diffusion probabilistic models (DDPM) [32], score-based generative models (SGMs) [33, 20], and stochastic differential equations (SDEs) [21, 34]. DDPM represents the forward and reverse process as two Markov chains, leveraging the Markov property to factorize the joint distribution into the product of transition kernels. SGMs introduce a sequence of intensifying Gaussian noise to perturb data, jointly estimating the score function for all noisy data distributions. Samples are generated by chaining the score functions at decreasing noise levels with score-based sampling methods such as Monte Carlo and Langevin Dynamics[35, 20]. SDEs perturb data to noise with a stochastic differential equation [21], whose forward process can be viewed as the continuous version of DDPM and SGM. Moreover, recent advances [25, 36, 22] have also enabled control over the generation process for conditional diffusion to generate specific images and text.

To our knowledge, recent studies [37–40] have explored integrating diffusion models into sequential recommendation. However, these approaches still adhere to the learning-to-classify paradigm, inevitably requiring negative sampling during training. For instance, Li et al. [37] and Du et al. [38] apply *softmax* cross-entropy loss [41] on the predicted logits of candidates, treating all non-target items as negative samples. While Wang et al. [39] uses binary cross-entropy loss, taking the next item as positive and randomly sampling a non-interacted item as the negative counterpart. They also incorporate contrastive loss for better classification, which requires substantial negative samples. However, diffusion model is mostly used for adding noise in the training samples for robustness, and the learning objectives are largely categorized as classification instead of generation. DiffRec by Wang et al. [40] proposes to apply diffusion on user's interaction vectors (*i.e.,* multi-hot vectors) for collaborative recommendation, where "1" indicates a positive interaction and "0" suggests a potential negative.

In contrast, our proposed DreamRec reshapes sequential recommendation as a learning-to-generate task. Specifically, DreamRec directly generates the oracle item tailored to user behavior sequence, transcending limitations of the concrete items in the candidate set and encouraging exploration of the underlying data distribution without the need of negative sampling.

## 3 Preliminary

We recap the basic notions of diffusion model, as established by the pioneering DDPM framework [32]. In this paper, we define the lower-case letter in bold as a variable, whose superscript refers to the diffusion step. We keep other notations the same as DDPM [32].

The fundamental objective of a generative model parameterized by $\theta$ is to model the underlying data-generation distribution, denoted by $p_\theta(\mathbf{x}^0)$, where $\mathbf{x}^0$ is the target variable. DDPM, a representative formulation of diffusion model, formulates two Markov chains, leveraging the chain rule of probabilities and the Markov property, to model the underlying distribution.

In the *forward (noising) process*, DDPM gradually adds Gaussian noise to $\mathbf{x}^0$ with a variance schedule $[\beta_1, \beta_2, \ldots, \beta_T]$:

$$q(\mathbf{x}^{1:T}|\mathbf{x}^0) = \prod_{t=1}^{T} q(\mathbf{x}^t|\mathbf{x}^{t-1}), \qquad q(\mathbf{x}^t|\mathbf{x}^{t-1}) = \mathcal{N}(\mathbf{x}^t; \sqrt{1-\beta_t}\mathbf{x}^{t-1}, \beta_t\mathbf{I}). \tag{1}$$

Let $\alpha_t = 1 - \beta_t$, $\bar{\alpha}_t = \prod_{s=1}^{t} \alpha_s$, $\boldsymbol{\epsilon} \sim \mathcal{N}(\mathbf{0}, \mathbf{I})$, we have $\mathbf{x}^t = \sqrt{\bar{\alpha}_t}\mathbf{x}^0 + \sqrt{1-\bar{\alpha}_t}\boldsymbol{\epsilon}$. DDPM jointly models the target variable $\mathbf{x}^0$ alongside a set of latent variables denoted by $\mathbf{x}^1, \ldots, \mathbf{x}^T$ as a Markov chain with Gaussian transitions:

$$p_\theta(\mathbf{x}^{0:T}) = p(\mathbf{x}^T) \prod_{t=1}^{T} p_\theta(\mathbf{x}^{t-1}|\mathbf{x}^t), \qquad p_\theta(\mathbf{x}^{t-1}|\mathbf{x}^t) = \mathcal{N}(\mathbf{x}^{t-1}; \boldsymbol{\mu}_\theta(\mathbf{x}^t, t), \boldsymbol{\Sigma}_\theta(\mathbf{x}^t, t)), \tag{2}$$

where the initial state is a Gaussian noise $\mathbf{x}^T \sim \mathcal{N}(\mathbf{0}, \mathbf{I})$. This is called the *reverse (denoising) process* of DDPM.

At the core of the generation task is to optimize the underlying data generating distribution $p_\theta(\mathbf{x}^0)$, which is performed by optimizing the variational bound of negative log-likelihood. In DDPM, this equals minimizing the KL divergence between $q(\mathbf{x}^{0:T})$ and $p_\theta(\mathbf{x}^{0:T})$:

$$\mathbb{E}\left[-\log p_\theta(\mathbf{x}^0)\right] \leq D_{KL}\left(q\left(\mathbf{x}_0, \mathbf{x}_1, \cdots, \mathbf{x}_T\right) \| p_\theta\left(\mathbf{x}_0, \mathbf{x}_1, \cdots, \mathbf{x}_T\right)\right) \tag{3}$$

$$= \mathbb{E}_{q(\mathbf{x}_0, \mathbf{x}_1, \cdots, \mathbf{x}_T)}\left[-\log p\left(\mathbf{x}_T\right) - \sum_{t=1}^{T} \log \frac{p_\theta\left(\mathbf{x}_{t-1} \mid \mathbf{x}_t\right)}{q\left(\mathbf{x}_t \mid \mathbf{x}_{t-1}\right)}\right] + C_1 \tag{4}$$

$$= \sum_{t=1}^{T} \underbrace{D_{KL}\left(q(\mathbf{x}^{t-1}|\mathbf{x}^t, \mathbf{x}^0)\|p_\theta(\mathbf{x}^{t-1}|\mathbf{x}^t)\right)}_{:=L_{t-1}} + C_2 \tag{5}$$

where $C_1$ and $C_2$ are constants that are independent of the model parameter $\theta$. Using Bayes' theorem, the posterior distribution $q(\mathbf{x}^{t-1}|\mathbf{x}^t, \mathbf{x}^0)$ could be solved in closed form:

$$q(\mathbf{x}^{t-1}|\mathbf{x}^t, \mathbf{x}^0) = \mathcal{N}(\mathbf{x}^{t-1}; \tilde{\boldsymbol{\mu}}_t(\mathbf{x}^t, \mathbf{x}^0), \tilde{\beta}_t\mathbf{I}), \tag{6}$$

where

$$\tilde{\boldsymbol{\mu}}_t(\mathbf{x}^t, \mathbf{x}^0) = \frac{\sqrt{\bar{\alpha}_{t-1}}\beta_t}{1-\bar{\alpha}_t}\mathbf{x}^0 + \frac{\sqrt{\alpha_t}(1-\bar{\alpha}_t)}{1-\bar{\alpha}_t}\mathbf{x}^t \quad \text{and} \quad \tilde{\beta}_t = \frac{1-\bar{\alpha}_{t-1}}{1-\bar{\alpha}_t}\beta_t. \tag{7}$$

Further reparameterizing $\boldsymbol{\mu}_\theta(\mathbf{x}^t, t)$ as:

$$\boldsymbol{\mu}_\theta(\mathbf{x}^t, t) = \frac{1}{\sqrt{\alpha_t}}\left(\mathbf{x}^t - \frac{1-\alpha_t}{\sqrt{1-\bar{\alpha}_t}}\boldsymbol{\epsilon}_\theta(\mathbf{x}^t, t)\right), \tag{8}$$

the $t$-th term of the training objective in Equation (5) is simplified to:

$$L_{t-1} = \mathbb{E}_{\mathbf{x}^0, \boldsymbol{\epsilon}}\left[\frac{\beta_t^2}{2\tilde{\beta}_t\alpha_t(1-\bar{\alpha}_t)}||\boldsymbol{\epsilon}_0 - \boldsymbol{\epsilon}_\theta(\sqrt{\bar{\alpha}_t}\mathbf{x}^0 + \sqrt{1-\bar{\alpha}_t}\boldsymbol{\epsilon}, t)||^2\right] + C. \tag{9}$$

Following DDPM [32], $\boldsymbol{\Sigma}_\theta(\mathbf{x}^t, t)$ is set to $\tilde{\beta}_t\mathbf{I}$ to match the variance of Equation (2) and Equation (6). The model architecture of $\boldsymbol{\epsilon}_\theta$ depends on specific tasks, such as U-Net for image generation [32] and Transformer for text generation [22].

# 4 Method

In this section, we elaborate on the proposed DreamRec, following the learning-to-generate paradigm with guided diffusion. We start by reformulating sequential recommendation as an oracle item generation task. Subsequently, we explain our approach to directly generate oracle item embeddings, taking inspiration from DDPM [32]. To conclude, we introduce our approach to enable personalized generation using classifier-free guidance.

## 4.1 Sequential Recommendation as Oracle Item Generation

Let $\mathcal{I}$ be the set of discrete items in the dataset. A historical interaction sequence is represented as $v_{1:n-1} = [v_1, v_2, \ldots, v_{n-1}]$, and $v_n$ is the subsequent consumed item of the sequence. Let $\mathcal{D} = \{[v_{1:n-1}, v_n]_m\}_{m=1}^{|\mathcal{D}|}$ stand for all the sequences within the training data, and $\mathcal{D}_t = \{[v_{1:n-1}]_m\}_{m=1}^{|\mathcal{D}_t|}$ denotes test sequences. Typically, each item $v \in \mathcal{I}$ is initially translated into its corresponding embedding vector $\mathbf{e}$. Thus, the interaction sequence can be represented as $\mathbf{e}_{1:n-1} = [\mathbf{e}_1, \mathbf{e}_2, \ldots, \mathbf{e}_{n-1}]$. The fundamental objective of sequential recommendation is to recommend the potential subsequent item that aligns with the user preference.

In this work, we propose DreamRec, a method that reshapes sequential recommendation as a learning-to-generate task. The principle behind it is that users tend to create an "oracle" item in their minds — an idealized item that they then search for in tangible form from the candidate set, when making a purchase. We assume that these oracle items are drawn from the same underlying distribution that generates observed items. In DreamRec, we model the underlying distribution as $p_\theta(\mathbf{e}_n^0|\mathbf{e}_{1:n-1})$ (*i.e.*, $p_\theta(\mathbf{e}_n|\mathbf{e}_{1:n-1})$), since the generation of next item $\mathbf{e}_n^0$ is highly related to historical interactions $\mathbf{e}_{1:n-1}$ in sequential recommendation. If $p_\theta(\mathbf{e}_n^0|\mathbf{e}_{1:n-1})$ can be precisely learned, it becomes possible to generate the oracle item from the interaction sequence through $p_\theta(\cdot|\mathbf{e}_{1:n-1})$. In DreamRec, we learn $p_\theta(\mathbf{e}_n^0|\mathbf{e}_{1:n-1})$ by employing a guided diffusion model.

## 4.2 Oracle Item Generation with Guided Diffusion

After framing sequential recommendation as an oracle item generation task, we proceed to introduce the learning and generation phases of DreamRec.

### 4.2.1 Learning Phase of DreamRec

A straightforward application of DDPM, as outlined in Section 3, cannot sufficiently achieve the oracle item generation objective in sequential recommendation. This is primarily because the denoising process modeled in Equation (2) lacks guidance from historical interactions, resulting in non-personalized generated items. To resolve this, we propose to guide the denoising process with the corresponding historical interaction sequence. Specifically, we first encode the interaction sequence $\mathbf{e}_{1:n-1} = [\mathbf{e}_1, \mathbf{e}_2, \ldots, \mathbf{e}_{n-1}]$ with a Transformer encoder:

$$\mathbf{c}_{n-1} = \textbf{T-enc}(\mathbf{e}_{1:n-1}). \quad (10)$$

This encoded interaction sequence, $\mathbf{c}_{n-1}$, conditions the denoising process as follows:

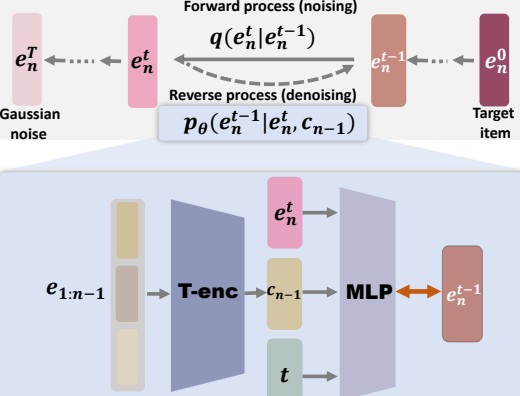

Figure 2: Framework of DreamRec. To achieve personalized denoising, DreamRec encodes historical interactions $\mathbf{e}_{1:n-1}$ to be guidance signal $\mathbf{c}_{n-1}$ with a Transformer encoder **T-enc**, subsequently utilizing $\mathbf{c}_{n-1}$ to guide the reverse process.

$$p_\theta(\mathbf{e}_n^{t-1}|\mathbf{e}_n^t, \mathbf{c}_{n-1}) = \mathcal{N}(\mathbf{e}_n^{t-1}; \boldsymbol{\mu}_\theta(\mathbf{e}_n^t, \mathbf{c}_{n-1}, t), \boldsymbol{\Sigma}_\theta(\mathbf{e}_n^t, \mathbf{c}_{n-1}, t)), \quad (11)$$

where the architecture of $\boldsymbol{\mu}_\theta(\mathbf{e}_n^t, \mathbf{c}_{n-1}, t)$ is an MLP in DreamRec depicted in Figure 2.

Similar to Equation (1), the forward process is formulated as a Markov chain of Gaussian transitions: $q(\mathbf{e}_n^t|\mathbf{e}_n^{t-1}) = \mathcal{N}(\mathbf{e}_n^t; \sqrt{1-\beta_t}\mathbf{e}_n^{t-1}, \beta_t\mathbf{I})$ with a variance schedule $[\beta_1, \beta_2, \ldots, \beta_T]$. Both the forward and reverse processes of DreamRec are also illustrated in Figure 2.

---

**Algorithm 1** Training phase of DreamRec

---

1: **repeat**
2:   $\mathbf{e}_n^0, \mathbf{e}_{1:n-1} \sim \mathcal{D}$                                         ▷ Sample and embed a data from training set.
3:   $\mathbf{c}_{n-1} = \text{T-enc}(\mathbf{e}_{1:n-1})$                                         ▷ Encode interaction sequence.
4:   With probability $p_u$:   $\mathbf{c}_{n-1} = \Phi$                     ▷ Perform unconditional training with probability $p_u$.
5:   $t \sim \text{Uniform}(\{1, \dots, T\})$                                         ▷ Sample diffusion step.
6:   $\epsilon \sim \mathcal{N}(\mathbf{0}, \mathbf{I})$                                         ▷ Sample Gaussian noise.
7:   $\mathbf{e}_n^t = \sqrt{\bar{\alpha}_t}\mathbf{e}_n^0 + \sqrt{1-\bar{\alpha}_t}\epsilon$                          ▷ Corrupt the traget item with Gaussian noise.
8:   $\theta = \theta - \mu\nabla_\theta \left\| \mathbf{e}_n^0 - f_\theta(\mathbf{e}_n^t, \mathbf{c}_{n-1}, t) \right\|^2$          ▷ Take gradient descent step, $\mu$ is the step size.
9: **until** converged

---

**Algorithm 2** Generation phase of DreamRec

---

1: $\mathbf{e}_{1:n-1} \sim \mathcal{D}_t$                                         ▷ Sample and embed a data from testing set.
2: $\mathbf{e}_n^T \sim \mathcal{N}(\mathbf{0}, \mathbf{I})$                                         ▷ Sample Gaussian noise.
3: $\mathbf{c}_{n-1} = \text{T-enc}(\mathbf{e}_{1:n-1})$                                         ▷ Encode interaction sequence.
4: **for** $t = T, \dots, 1$ **do**                                         ▷ Denoise for T steps.
5:   $\mathbf{z} \sim \mathcal{N}(\mathbf{0}, \mathbf{I})$ if $t > 1$, else $\mathbf{z} = \mathbf{0}$                ▷ Sample denoising variance.
6:   $\tilde{f}_\theta(\mathbf{e}_n^t, \mathbf{c}_{n-1}, t) = (1+w)\, f_\theta(\mathbf{e}_n^t, \mathbf{c}_{n-1}, t) - w\, f_\theta(\mathbf{e}_n^t, \Phi, t)$          ▷ Control the strength of guidance.
7:   $\mathbf{e}_n^{t-1} = \frac{\sqrt{\bar{\alpha}_{t-1}}\beta_t}{1-\bar{\alpha}_t}\tilde{f}_\theta(\mathbf{e}_n^t, \mathbf{c}_{n-1}, t) + \frac{\sqrt{\alpha_t}(1-\bar{\alpha}_{t-1})}{1-\bar{\alpha}_t}\mathbf{e}_n^t + \sqrt{\tilde{\beta}_t}\mathbf{z}$          ▷ Denoise for one step.
8: **end for**
9: **return** $\mathbf{e}_n^0$

---

Thereafter, DreamRec improves upon $L_{t-1}$ of Equation (5) with guidance signal, and achieves the learning objective:

$$L_{t-1} = D_{KL}(q(\mathbf{e}_n^{t-1}|\mathbf{e}_n^t, \mathbf{e}_{n-1}^0)||p_\theta(\mathbf{e}_n^{t-1}|\mathbf{e}_n^t, \mathbf{c}_{n-1})). \tag{12}$$

We employ another reparameterization that predicts target sample $\mathbf{e}_n^0$ rather than the added noise $\epsilon$ as in Equation (8)[3]:

$$\boldsymbol{\mu}_\theta(\mathbf{e}_n^t, \mathbf{c}_{n-1}, t) = \sqrt{\bar{\alpha}_{t-1}}f_\theta(\mathbf{e}_n^t, \mathbf{c}_{n-1}, t) + \frac{\sqrt{\alpha_t}(1-\bar{\alpha}_{t-1})}{\sqrt{1-\bar{\alpha}_t}}\epsilon, \tag{13}$$

and Equation (12) converts to another format:

$$L_{t-1} = \mathbb{E}_{\mathbf{e}_n^0, \epsilon}\left[\frac{\bar{\alpha}_{t-1}}{2\tilde{\beta}_t}||\mathbf{e}_n^0 - f_\theta(\sqrt{\bar{\alpha}_t}\mathbf{e}_n^0 + \sqrt{1-\bar{\alpha}_t}\epsilon, \mathbf{c}_{n-1}, t)||^2\right] + C. \tag{14}$$

Besides the conditional diffusion model, guided diffusion typically necessitates an additional unconditional one [25], which can be jointly trained using a *classifier-free guidance* scheme [36]. Specifically, during training, we randomly replace guidance signal $\mathbf{c}_{n-1}$ by a dummy token $\Phi$ with probability $p_u$ to achieve the training of unconditional diffusion model.

It is important to note that in Equation (14), $\mathbf{e}_n^0$ denotes the observed target item in interaction sequence, and $\epsilon$ denotes the Gaussian noise. Therefore, DreamRec's training procedure does not require negative sampling. Instead, it concentrates on recovering the target item in the interaction sequence. Algorithm 1 shows the details of DreamRec's training phase.

#### 4.2.2 Generation Phase of DreamRec

In the generation phase, we target to generate personalized oracle items for different users, given their historical interactions. To manipulate the influence of the guidance signal $\mathbf{c}_{n-1}$, we would modify $f_\theta(\mathbf{e}_n^t, \mathbf{c}_{n-1}, t)$ to conform to the following format:

$$\tilde{f}_\theta(\mathbf{e}_n^t, \mathbf{c}_{n-1}, t) = (1+w)\, f_\theta(\mathbf{e}_n^t, \mathbf{c}_{n-1}, t) - w\, f_\theta(\mathbf{e}_n^t, \Phi, t), \tag{15}$$

---

[3]Please note that predicting $\mathbf{e}_n^0$ and predicting $\epsilon$ are equivalent due to $\mathbf{e}_n^t = \sqrt{\bar{\alpha}_t}\mathbf{e}_n^0 + \sqrt{1-\bar{\alpha}_t}\epsilon$. Refer to Appendix A for more details.

where $w$ is a hyperparameter controlling the strength of $c_{n-1}$: A higher $w$ value can enhance personalized guidance, but it could potentially undermine diffusion generalization, consequently deteriorating the quality of the generated oracle item.

Based on Equation (7), the one-step denoising process is straightforward as follows:

$$\mathbf{e}_n^{t-1} = \frac{\sqrt{\bar{\alpha}_{t-1}}\beta_t}{1-\bar{\alpha}_t}\tilde{f}_\theta(\mathbf{e}_n^t, \mathbf{c}_{n-1}, t) + \frac{\sqrt{\alpha_t}(1-\bar{\alpha}_{t-1})}{1-\bar{\alpha}_t}\mathbf{e}_n^t + \sqrt{\tilde{\beta}_t}\mathbf{z}, \quad \mathbf{z} \sim \mathcal{N}(\mathbf{0}, \mathbf{I}). \quad (16)$$

During the inference stage, for a user characterized by historical interactions encoded as $\mathbf{c}_{n-1}$, the oracle item $\mathbf{e}_n^0$ can be generated by denoising a Gaussian sample $\mathbf{e}_n^T \sim \mathcal{N}(\mathbf{0}, \mathbf{I})$ for $T$ steps, in accordance with Equation (16). The generation phase of DreamRec is demonstrated in Algorithm 2.

### 4.2.3 Retrieval of Recommendation List

After generating the oracle item, the subsequent step involves obtaining the recommendation list tailored to the specific user. To achieve this, we retrieve the K-nearest items to the oracle item within the candidate set using an inner product measurement, forming the recommendaiton list. It's crucial to emphasize that the selection of K-nearest items is exclusive to the retrieval of the recommendation list and does not figure into the training phase. Conceptually, DreamRec transcends the confines of a limited candidate set, venturing into the entire item space in its pursuit of the oracle item.

## 5 Experiments

In this section, we conduct experiments to demonstrate that: 1) DreamRec provides a powerful learning-to-generate framework for sequential recommendation; 2) DreamRec can better explore the item space without negative sampling; and 3) Guiding the diffusion process with historical interactions is important for personalized oracle item generation in DreamRec.

### 5.1 Experimental Settings

**Datasets.** We use three datasets from real-world sequential recommendation scenarios: YooChoose, KuaiRec, and Zhihu (the statistics of datasets are illustrated in Appendix B):

- **YooChoose** dataset comes from RecSys Challenge 2015 [4]. We preserve the purchase sequences for a moderate size of data. We only retain items with at least 5 interactions to avoid cold-start issue. Additionally, we exclude sequences that are shorter than 3 interactions in length.

- **KuaiRec** [42] dataset is collected from the recommendation logs of a video-sharing mobile app. We also remove items that are interacted with less than 5 times and sequences shorter than 3.

- **Zhihu** [43] dataset is collected from a socialized knowledge-sharing community. Users are presented with a recommended Q&A list and they can read their preferred ones. We remove items that are read less than 5 times and sequences that are shorter than 3 in length.

For all datasets, we first sort all sequences in chronological order, and then split the data into training, validation and testing data at the ratio of 8:1:1.

**Baselines.** We compare DreamRec against several competitive models, including GRU4Rec [4], Caser [10], SASRec [5], S-IPS [7], AdaRanker [12], CL4SRec [8] and DiffRec [40]. GRU4Rec, Caser, and SASRec adopt recurrent neural networks, convolutional neural networks, and Transformer encoders respectively to capture sequential patterns of user behaviors. S-IPS and AdaRanker leverage causal inference and neural process to address the issues of selection bias and temporal dynamics in sequential recommendation. CL4SRec designs three data augmentation methods and applies contrastive learning techniques to enhance the classification ability of sequential recommender. DiffRec proposes to incorporate the diffusion model in collaborative filtering, but it applies diffusion on user's interaction vectors — multi-hot vectors whose element of "1" indicates a positive interaction while "0" suggests a potential negative.

**Training Protocol.** We implement all models with Python 3.7 and PyTorch 1.12.1 in Nvidia GeForce RTX 3090. We preserve the last 10 interactions as historical sequence. For sequences with less than

---

[4] https://recsys.acm.org/recsys15/challenge/

Table 1: Overall performance comparison. The boldface denotes the best performance while the underline indicates the second best. The experiments are conducted 5 times and the average and standard deviation are reported.

| | YooChoose | | KuaiRec | | Zhihu | |
|---|---|---|---|---|---|---|
| | HR@20(%) | NDCG@20(%) | HR@20(%) | NDCG@20(%) | HR@20(%) | NDCG@20(%) |
| GRU4Rec | 3.89±0.11 | 1.62±0.02 | 3.32±0.11 | 1.23±0.08 | 1.78±0.12 | 0.67±0.03 |
| Caser | 4.06±0.12 | 1.88±0.09 | 2.88±0.19 | 1.07±0.07 | 1.57±0.05 | 0.59±0.01 |
| SASRec | 3.68±0.08 | 1.63±002 | 3.92±0.18 | 1.53±0.11 | 1.62±0.01 | 0.60±0.03 |
| IPS | 3.81±0.05 | 1.73±0.03 | 3.73±0.03 | 1.40±0.05 | 1.66±0.04 | 0.64±0.02 |
| AdaRanker | 3.74±0.06 | 1.67±0.04 | 4.14±0.09 | 1.89±0.05 | 1.70±0.04 | 0.61±0.02 |
| CL4SRec | 4.45±0.04 | 1.86±0.02 | 4.25±0.10 | 2.01±0.09 | 2.03±0.06 | 0.74±0.03 |
| DiffRec | 4.33±0.02 | 1.84±0.01 | 3.74±0.08 | 1.77±0.05 | 1.82±0.03 | 0.65±0.09 |
| DreamRec | **4.78**±0.06 | **2.23**±0.02 | **5.16**±0.05 | **4.11**±0.02 | **2.26**±0.07 | **0.79**±0.01 |

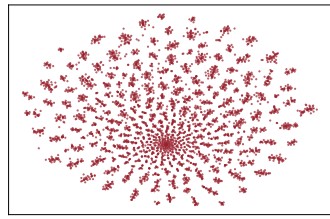 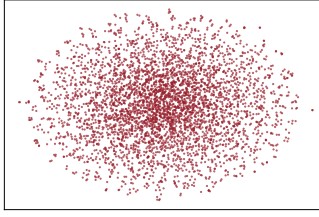 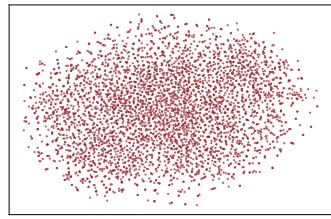

(a) SASRec (no negative sampling).     (b) SASRec.     (c) DreamRec.

Figure 3: Visualization of the learned item embeddings on Zhihu dataset using T-SNE, where red points represent items. Figure 3a shows that without negative sampling, the item embeddings of SASRec are crowded in limited discrete zones. Figure 3b shows that With negative sampling, the item embeddings of SASRec concentrate on only part of the item space. Figure 3c shows that DreamRec explores most of the item space without requiring negative sampling.

10 interactions, we would pad them to 10 with a padding token. We leverage AdamW as the optimizer. The embedding dimension of items is fixed as 64 across all models. The learning rate is tuned in the range of [0.01, 0.005, 0.001, 0.0005, 0.0001, 0.00005]. Despite that DreamRec does not require L2 regularization, we tune the weight of L2 regularization for all baselines in the range of [1e-3, 1e-4, 1e-5, 1e-6, 1e-7]. For all baselines, we conduct negative sampling from the uniform distribution at the ratio of 1: 1, which is not conducted in DreamRec. For our DreamRec, we fix the unconditional training probability $p_u$ as 0.1 suggested by [36]. We search the total diffusion step $T$ in the range of [50, 100, 200, 500, 1000, 2000], and the personalized guidance strength $w$ in the range of [0, 2, 4, 6, 8, 10].

**Evaluation Protocol.** We follow the widely used top-K protocol to evaluate the performance of sequential recommendation and adopt two widely used metrics: hit ratio (HR) and normalized discounted cumulative gain (NDCG) [4, 5]. Sequential recommenders within the learning-to-classify paradigm usually leverage the classification logits on candidate items to select top-K items. For DreamRec, we first generate the oracle item with Algorithm 2, and then we find the K-nearest items in the candidate set for top-K recommendation with the measurement of inner product. Note that the selection of K-nearest items is only conducted at the evaluation phase, and is not involved in the training phase.

## 5.2 Main Results

In this section, we compare DreamRec against baseline models in terms of top-K recommendation performance. Table 1 shows the experimental results. Overall, DreamRec substantially and consistently outperforms compared models, which demonstrates the superiority of our proposed learn-to-generate paradigm in sequential recommendation.

Note that all baselines are within the learning-to-classify paradigm: GRU4Rec, Caser and SASRec are backbone models; IPS and AdaRanker enhance classification ability by solving data biasing issues; and CL4SRec further equips classification objectives with contrastive learning techniques. We do witness in Table 1 that these auxiliary tasks improve the performance of backbone models.

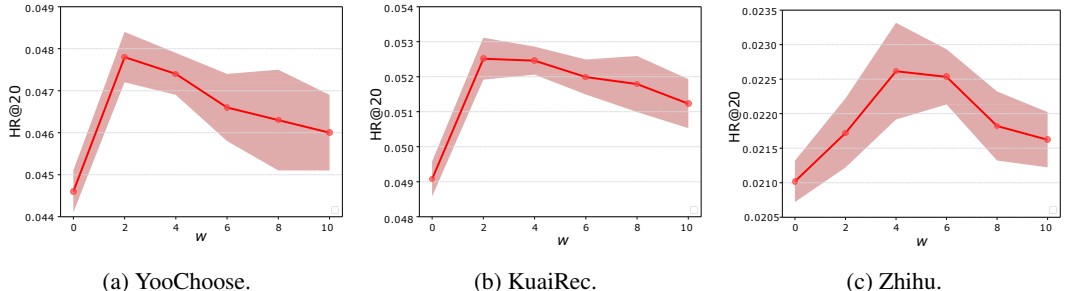

|  |  |  |
|:---:|:---:|:---:|
| (a) YooChoose. | (b) KuaiRec. | (c) Zhihu. |

Figure 4: Ablation study of classifier-free guidance by adjusting the guidance strength $w$ on Yoo-Choose (subfigure 4a), KuaiRec (subfigure 4b) , and Zhihu (subfigure 4c) datasets.

In terms of DreamRec, it reshapes sequential recommendation purely as an oracle item generation task without those auxiliary tasks, and it has already achieved better performance in our experiments. We believe designing auxiliary tasks can also boost the generation of oracle items in sequential recommendation, yet we leave this as future work since it is beyond the scope of this paper.

### 5.3 Visualization

In this section, we visualize the learned item embeddings using T-SNE [44] to demonstrate that DreamRec can well explore the underlying distribution of item space without negative sampling.

Specifically, we first train three recommenders on Zhihu dataset (results on other datasets included in Appendix C): 1) SASRec without negative sampling; 2) SASRec with negative sampling; and 3) DreamRec that does not require negative sampling. Then we use T-SNE to visualize the learned item embeddings under the default setting of scikit-learn [5]. Figure 3 demonstrates the visualization results, where each item is represented as a red point. From Figure 3 we can observe that: 1) Negative sampling is necessary for classification-based sequential recommenders: if no negative sampling is conducted, plenty of items may be crowded pathologically in limited discrete zones of the item space, making them indistinguishable; 2) Negative sampling makes classification easier since items are better shattered, but they are biased to concentrate more on part of the zone of item space; and 3) By reshaping sequential recommendation as a learning-to-generate framework, DreamRec explores most of the zones of item space without requiring negative sampling. These observations provide strong empirical support for our claims made in Section 1.

### 5.4 Controlling the Personalized Guidance of DreamRec

As introduced in Section 4.2.2, achieving personalized oracle item generation requires guidance from the corresponding interaction sequence. To achieve controllable guidance in DreamRec, we adopt the classifier-free guidance technique. However, we should carefully adjust the value of guidance strength $w$ in Equation (15), as a higher value may hurt the generalization of diffusion and lead to generating lower-quality oracle items. To shed light on this issue, we conduct experiments on the three datasets by adjusting the value of $w$. As shown in Figure 4, we observed that increasing the value of $w$ initially leads to better recommendation accuracy. This supports our intuition that stronger guidance leads to better personalization. However, as we continue to increase the value of $w$, we observe a decline in performance. This is in line with our analysis that concentrating too much on the guidance signal may hurt the generation quality of oracle items.

## 6   Conclusion and Limitations

We propose DreamRec, reshaping sequential recommendation as a learning-to-generate task, instead of a learning-to-classify task as almost all previous methods do. DreamRec is based on the analysis of user behaviors that, after several interactions with the recommendation system, users tend to fantasize about an oracle item they would "ideally" consume. The oracle item does not have to be the next item in the dataset, and even should not be limited to the pre-defined candidate set. By

---

[5]https://scikit-learn.org/stable/modules/generated/sklearn.manifold.TSNE.html

modeling the underlying data-generation distribution with diffusion model, DreamRec promises to generate unobserved oracle items. Moreover, targeting to model the data-generation process directly, DreamRec makes it possible to discard negative samples in sequential recommendation, which can hardly be achieved by previous classification-based models. Experiments show that DreamRec brings consistent improvements to sequential recommendation, implying its superiority in modeling user behaviors.

Meanwhile, DreamRec also has a few limitations: 1) the sampling process is quite slow with the iteration of total diffusion steps; and 2) the training process is more time-consuming. We believe these can be resolved in further research with more advanced generation models such as consistency model [45]. Moreover, as an initial attempt of reshaping sequential recommendation as an oracle item generation task, DreamRec provides many research opportunities such as designing auxiliary tasks (*e.g.,* contrastive learning or data augmentation) to enhance oracle item generation. Another research line could be designing other encoders for guidance representation, or other model architectures for the denoising process of DreamRec.

## 7 Broader Impact

The proposed DreamRec can significantly improve the performance of sequential recommendation. Therefore, it can be applied to real-world platforms to provide more satisfying recommendation results. One concern of generating oracle items is the potential for privacy disclosure. Despite that we encode the oracle item with vector representations, one may decode the representation and snoop on users' preference explicitly. Therefore, we kindly advise researchers to be cautious about the usage of generated oracle items.

## Acknowledgements

This research is supported by the National Natural Science Foundation of China (9227010114, U21B2026, 62302321), the University Synergy Innovation Program of Anhui Province (GXXT-2022-040), and the Hong Kong Polytechnic University under grant (P0045485).

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

# A  Proving the Equivalent of Equation (9) and (14)

Recall that in Equation (3) - (5), we have formulated the generation task in DDPM as optimizing the variational bound of negative log-likelihood, where the $t$-th term is as follows:

$$L_{t-1} = D_{KL}\left(q(\mathbf{x}^{t-1}|\mathbf{x}^t, \mathbf{x}^0)||p_\theta(\mathbf{x}^{t-1}|\mathbf{x}^t)\right). \tag{17}$$

Since $q(\mathbf{x}^{t-1}|\mathbf{x}^t, \mathbf{x}^0)$ and $p_\theta(\mathbf{x}^{t-1}|\mathbf{x}^t)$ are both Gaussian distributions, we can apply the Rao-Blackwellized Theorem [32] to compute Equation (17), resulting in:

$$L_{t-1} = \mathbb{E}_q\left[\frac{1}{2\tilde{\beta}_t}||\tilde{\boldsymbol{\mu}}_t(\mathbf{x}^t, \mathbf{x}^0) - \boldsymbol{\mu}_\theta(\mathbf{x}^t, t)||^2\right] + C, \tag{18}$$

where:

$$\tilde{\boldsymbol{\mu}}_t(\mathbf{x}^t, \mathbf{x}^0) = \frac{\sqrt{\bar{\alpha}_{t-1}}\beta_t}{1-\bar{\alpha}_t}\mathbf{x}^0 + \frac{\sqrt{\alpha_t}(1-\bar{\alpha}_{t-1})}{1-\bar{\alpha}_t}\mathbf{x}^t. \tag{19}$$

Note that $\mathbf{x}^t = \sqrt{\bar{\alpha}_t}\mathbf{x}^0 + \sqrt{1-\bar{\alpha}_t}\boldsymbol{\epsilon}$, we have:

$$
\begin{aligned}
\tilde{\boldsymbol{\mu}}_t(\mathbf{x}^t, \mathbf{x}^0) &= \frac{\sqrt{\bar{\alpha}_{t-1}}\beta_t}{1-\bar{\alpha}_t}\mathbf{x}^0 + \frac{\sqrt{\alpha_t}(1-\bar{\alpha}_{t-1})}{1-\bar{\alpha}_t}\mathbf{x}^t \\
&= \frac{\sqrt{\bar{\alpha}_{t-1}}\beta_t}{1-\bar{\alpha}_t}\frac{\mathbf{x}^t - \sqrt{1-\bar{\alpha}_t}\boldsymbol{\epsilon}}{\sqrt{\bar{\alpha}_t}} + \frac{\sqrt{\alpha_t}(1-\bar{\alpha}_{t-1})}{1-\bar{\alpha}_t}\mathbf{x}^t \\
&= \frac{1}{\sqrt{\alpha_t}}\left(\mathbf{x}_t - \frac{1-\alpha_t}{\sqrt{1-\bar{\alpha}_t}}\boldsymbol{\epsilon}\right).
\end{aligned}
\tag{20}
$$

In DDPM, $\boldsymbol{\mu}_\theta$ is reparametrized as:

$$\boldsymbol{\mu}_\theta(\mathbf{x}^t, t) = \frac{1}{\sqrt{\alpha_t}}\left(\mathbf{x}^t - \frac{1-\alpha_t}{\sqrt{1-\bar{\alpha}_t}}\boldsymbol{\epsilon}_\theta(\mathbf{x}^t, t)\right). \tag{21}$$

Substituting Equation (20) and (21) back to Equation (18), we can obtain the simplified version of $L_{t-1}$ as expressed in Equation (9):

$$L_{t-1} = \mathbb{E}_{\mathbf{x}^0,\boldsymbol{\epsilon}}\left[\frac{\beta_t^2}{2\tilde{\beta}_t\alpha_t(1-\bar{\alpha}_t)}||\boldsymbol{\epsilon} - \boldsymbol{\epsilon}_\theta(\sqrt{\bar{\alpha}_t}\mathbf{x}^0 + \sqrt{1-\bar{\alpha}_t}\boldsymbol{\epsilon}, t)||^2\right] + C. \tag{22}$$

We can adopt an alternative reparametrization of $\boldsymbol{\mu}_\theta$ similar to Equation (13):

$$\boldsymbol{\mu}_\theta(\mathbf{x}^t, t) = \sqrt{\bar{\alpha}_{t-1}}f_\theta(\mathbf{x}^t, t) + \frac{\sqrt{\alpha_t}(1-\bar{\alpha}_{t-1})}{\sqrt{1-\bar{\alpha}_t}}\boldsymbol{\epsilon}. \tag{23}$$

Then we can derive:

$$
\begin{aligned}
&\tilde{\boldsymbol{\mu}}_t(\mathbf{x}^t, \mathbf{x}^0) - \boldsymbol{\mu}_\theta(\mathbf{x}^t, t) \\
&= \frac{1}{\sqrt{\alpha_t}}\left(\mathbf{x}^t - \frac{1-\alpha_t}{\sqrt{1-\bar{\alpha}_t}}\boldsymbol{\epsilon}\right) - \sqrt{\bar{\alpha}_{t-1}}f_\theta(\mathbf{x}^t, t) - \frac{\sqrt{\alpha_t}(1-\bar{\alpha}_{t-1})}{\sqrt{1-\bar{\alpha}_t}}\boldsymbol{\epsilon} \\
&= \frac{1}{\sqrt{\alpha_t}}\left(\sqrt{\bar{\alpha}_t}\mathbf{x}^0 + \sqrt{1-\bar{\alpha}_t}\boldsymbol{\epsilon} - \frac{1-\alpha_t}{\sqrt{1-\bar{\alpha}_t}}\boldsymbol{\epsilon}\right) - \sqrt{\bar{\alpha}_{t-1}}f_\theta(\mathbf{x}^t, t) - \frac{\sqrt{\alpha_t}(1-\bar{\alpha}_{t-1})}{\sqrt{1-\bar{\alpha}_t}}\boldsymbol{\epsilon} \\
&= \sqrt{\bar{\alpha}_{t-1}}\mathbf{x}^0 + \frac{\sqrt{\alpha_t}(1-\bar{\alpha}_{t-1})}{\sqrt{1-\bar{\alpha}_t}}\boldsymbol{\epsilon} - \sqrt{\bar{\alpha}_{t-1}}f_\theta(\mathbf{x}^t, t) - \frac{\sqrt{\alpha_t}(1-\bar{\alpha}_{t-1})}{\sqrt{1-\bar{\alpha}_t}}\boldsymbol{\epsilon} \\
&= \sqrt{\bar{\alpha}_{t-1}}(\mathbf{x}^0 - f_\theta(\mathbf{x}^t, t)).
\end{aligned}
\tag{24}
$$

Therefore Equation (18) can be simplified to another version:

$$L_{t-1} = \mathbb{E}_{\mathbf{x}^0,\boldsymbol{\epsilon}}\left[\frac{\bar{\alpha}_{t-1}}{2\tilde{\beta}_t}||\mathbf{x}^0 - f_\theta(\sqrt{\bar{\alpha}_t}\mathbf{x}^0 + \sqrt{1-\bar{\alpha}_t}\boldsymbol{\epsilon}, t)||^2\right] + C \tag{25}$$

So far we have proven that Equation (22) (predicting noise) and (25) (predicting target) are two equivalent versions of training objective for diffusion in theory. In practice, DDPM shows that Equation (22) performs well in image generation, while study [22] shows Equation (25) in more suitable in text generation. In sequential recommendation, we adopt Equation (25) (or equivalently, Equation (14)) to acquire our objective in DreamRec for oracle item generation.

# B  Detailed Experimental Settings

## B.1  Statistics of Datasets

The statistics of the adopted datasets are summarized in Table 2.

Table 2: Statistics of datasets.

| Dataset | YooChoose | KuaiRec | Zhihu |
|---|---|---|---|
| #sequences | 128,468 | 92,090 | 11,714 |
| #items | 9,514 | 7,261 | 4,838 |
| #interactions | 539,436 | 737,163 | 77,712 |

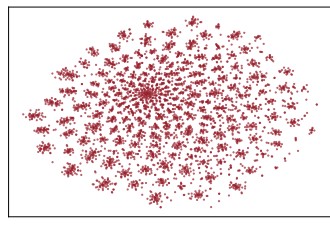 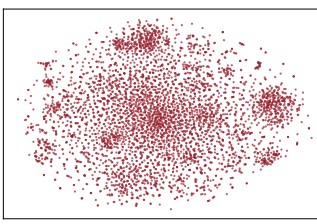 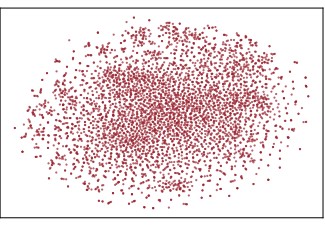

(a) SASRec (no negative sampling).          (b) SASRec.          (c) DreamRec.

Figure 5: Visualization of the learned item embeddings on YooChoose dataset using T-SNE. Dream-Rec explores most of the item space without requiring negative sampling.

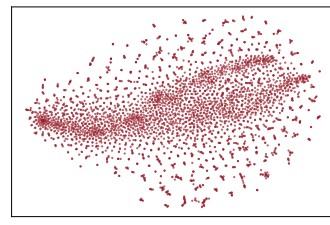 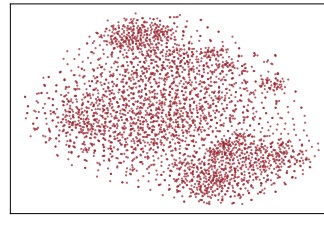 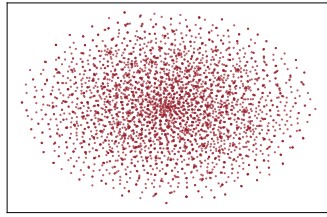

(a) SASRec (no negative sampling).          (b) SASRec.          (c) DreamRec.

Figure 6: Visualization of the learned item embeddings on KuaiRec dataset using T-SNE. DreamRec explores more of the item space without requiring negative sampling.

# C  More Ablation Studies

## C.1  Visualization on YooChoose and KuaiRec Datasets.

We further visualize the learned item embedding visualization on YooChoose and KuaiRec datasets in Figure 5 and Figure 6. Similar to Zhihu dataset (Figure 3), SASRec without negative sampling may fail to distinguish different items, since plenty of items gather in limited zones of the item space. Consequently, negative sampling is necessary for classification-based sequential recommenders. In contrast, DreamRec directly models the data-generation distribution and can better explore item space without the requirement of negative sampling.

