# OpenReview forum: "Generate What You Prefer: Reshaping Sequential Recommendation via Guided Diffusion"
_NeurIPS.cc/2023/Conference — NeurIPS 2023 poster_

### Official Review · Reviewer_b6yV · 2023-07-05

**Soundness:** 3 good
**Presentation:** 3 good
**Contribution:** 3 good
**Rating:** 6
**Confidence:** 3

**Summary:**

This paper presents an innovative paradigm shift in sequential recommendation systems. It departs from the traditional learning-to-classify approach, which includes negative sampling, to a learning-to-generate model, proposing a novel system called Diff4Rec, grounded on guided diffusion. This approach is predicated on the observation that users imagine an ideal or "oracle" item after several interactions with a recommendation system.

The authors posit that Diff4Rec can generate these oracle items, modeling the underlying item-generation distribution through a diffusion process, and is not limited to a predefined candidate set. This approach allows for discarding negative samples, which previous models could not accomplish, as it models the data-generation process directly.

The paper suggests that Diff4Rec effectively mirrors human behavior better than previous systems by capturing a user's preference more accurately and directly, avoiding the noisy or easy supervisions from negative samples. The authors evaluate Diff4Rec through various experiments, where it demonstrates consistent improvements over existing sequential recommendation models. In conclusion, this paper introduces a transformative methodology in sequential recommendation, potentially broadening the capabilities of recommendation systems.

**Strengths:**

1. Raise a new learning-to-generate paradigm for recommendation using relative new diffusion model approach.
2. Good reproducibility: Code is released for readers; Hyperparameter settings are available.
3. Good to explain what each step is doing in the Algorithms.
4. Improvements over the baseline appear significant. But statistical significance analysis is missing.

**Weaknesses:**

1. In terms of recommendation, an important and indispensable step is not discussed and stated in the algorithm. After an oracle item is generated, to fetch one / or some items to recommend to users, we need to find the top nearest items to the oracle item in the embedding space. Though this step is described in the Experiment, it is fully ignored in the Method section. I understand the Method part described mainly the learning-to-generate paradigm. But this step is needed to complete the recommendation task. Moreover, to retrieve top nearest items, how to encode properly the embedding space for items becomes also a relevant question.
2. Might be good to list out the main contributions of the work.
3. Please avoid abusing the footnotes. Footnotes are good for external links and websites. If you do think footnotes are providing important information, please try to integrate them in the main text instead, e.g. 3 and 4. And try to avoid the controversial remarks irrelevant to the main contents, like 2: people working on learning to rank may not fully agree on 2.
4. In table 1, if authors meant to have %, better to add (%) after HR@20 and NDCG@20. It’s easy to be confused, even when I read the caption.

**Questions:**

1. A key success of the diffusion models in image generation is that it can encode some of undetermined features in the latent space so that random initializer helps to randomly sample those features in additional to the guidance signals. This implies, to a given user and interaction history, we can always generate different oracle items from different random seeds. How will this help on top item retrieval? For example, we can generate different oracle items and make a pooling to get an ensemble of oracles and then retrieve from that. How will this be different from retrieving one for each generated oracle? It would be very interesting for users to investigate on this end.
2. Can authors add one / more of the other diffusion model based recommendation model (19,20,21,36) results as a baseline? Tells us better the effect of with/without negative sampling.

**Limitations:**

1. Authors may disagree on whether selecting the nearest items on the embedding space is a necessary step for recommendation. I could also imagine a recommendation with a generative item, for example, a LLM decoder to generate recommended results from the oracle item embedding generated with the diffusion result. Maybe authors could discuss how to fetch / generate the recommended item(s) from the oracle item in the main text.

---

> ### Author Rebuttal · Authors · 2023-08-10
>
> > 1. *The step to retrive recommendation list is needed to complete the recommendation task. Moreover, to retrieve top nearest items, how to encode properly the embedding space for items becomes also a relevant question.*
>
>    Thank you for highlighting this point. We recognize the importance of identifying the top nearest items to the oracle item in recommendation tasks. We regret the oversight of detailing this in the Experiment section instead of the Method section. In the revision, we will introduce Section 4.2.3, titled "Retrieval of Recommendation Results," to address this aspect. We hope this addition clarifies your concerns.
>
>    Regarding the encoding of the item embedding space, we utilize a Transformer encoder in Diff4Rec to process item sequences. We concur that the choice of encoding strategy for this space is pivotal. To explore alternatives, we conducted experiments replacing the Transformer encoder with a GRU, with the results presented below:
>
> || Yoochoose|Yoochoose| KuaiRec| KuaiRec|Zhihu|Zhihu
> :--:|:--:|:--:|:--:|:--:|:--:|:--:
> ||HR@20(%) | NDCG@20(%)|HR@20(%)|NDCG@20(%)|HR@20(%)|NDCG@20(%)
> Transformer encoder|4.78|  2.23 |5.26|4.11|2.26|0.79
> GRU encoder|4.48|1.92|5.58|5.17|2.05|0.73
>
> ---
>
> > 2. *Might be good to list out the main contributions of the work.*
>
>    Thank you for this advice.  We will list out the main contributions in the revised version, focusing on: (1) Diff4Rec reshapes sequential recommendation as an oracle item generation task, (2) Diff4Rec does not require negative samples, since it explores the underlying distribution of observed interactions with diffusion model, and (3) We conduct experiments on three datasets to show the effectiveness of Diff4Rec. We believe this makes the presentation of our work more clear!
>
> ---
>
> > 3. *Please avoid abusing the footnotes. *
>
>    We apologize for the abuse of footnotes. We will integrate the key point of Footnote 3 and 4 into the main text in the revised version.
> We understand Footnote 2 is controversial since learning-to-rank has difference with learning-to-classify. What we would like to convey was to highlight their similarity in requiring negative samples for recommendation tasks.  In the revision, we will clarify their similarity more precisely in the main text, avoiding any controversial statements.  Thanks again for reminding us of this issue.
>
> ---
>
> > 4. *better to add (%) after HR@20 and NDCG@20.*
>
>    Thank you for this advice.  We will add (%) after HR@20 and NDCG@20 in Table 1 in the revised version.
>
> ---
>
> > 5. *For a given user and interaction history, we can always generate different oracle items from different random seeds. How will this help on top item retrieval? For example, we can generate different oracle items and make a pooling to get an ensemble of oracles and then retrieve from that. How will this be different from retrieving one for each generated oracle? It would be very interesting for users to investigate on this end.*
>
>    Thank you for this question.  We feel that this question is more about the understanding of diffusion model, and we are glad to share our understanding of it.
>
>    In image generation, if guidance signals are involved,  the generated images may be different, but they would center around the guidance signal. Even though they are different in RGB, it is hard to say that they are always different in certain latent spaces with certain measurements. In general, the generation process of diffusion is about removing certain noise from a pure Gaussian sample, and the guidance signal provides more delicate information about which part of noise to remove. Back to Diff4Rec, the generated oracle items can be different with different random seeds, but they are all recovered from Gaussian noises with the guidance of the same behavior sequence. Therefore, they could be similar in certain latent spaces with certain measurements, which, we believe, is an open scientific topic to explore.
>
>    Moreover, we do find it insightful to generate different oracle items and make a pooling to get an ensemble of oracles and then retrieve from that. We conduct mean pooling of 5 different oracle items, and the result is shown as follows:
>
> || Yoochoose|Yoochoose| KuaiRec| KuaiRec|Zhihu|Zhihu
> :--:|:--:|:--:|:--:|:--:|:--:|:--:
> ||HR@20(%) | NDCG@20(%)|HR@20(%)|NDCG@20(%)|HR@20(%)|NDCG@20(%)
> Diff4Rec|4.78|2.23|5.26|4.11| 2.26|0.79|
> ensemble|4.83|2.41|5.47|4.32|2.32|0.78|
>
>    The reason may be that the ensemble of oracles can effectively reduce the variance.
>
> ---
>
> > 6. *Can authors add one / more of the other diffusion  based recommendation model (19,20,21,36) results as a baseline?*
>
>    Thank you for the suggestions. We have added models from references [19], [20], and [36] as baselines for comparison in Table 4 of the PDF. Of these works, only [36] provides open-sourced code that allows direct reproduction. For [19] and [20], we carefully implemented their methods based on the paper's details to enable fair comparisons. We are still working to reproduce the approach [21] and will include it in the subsequent stages.
>
> ---
>
> > 7. *A  LLM decoder could generate recommended results from the oracle item embedding. Maybe authors could discuss how to fetch / generate the recommended item(s) from the oracle item in the main text.*
>
>    Thank you for this insightful comment! After analyzing this comment, we do believe it is more elegant to decode the oracle items explicitly, possibly by LLMs.
>    We do some initial attempts and would like to finetune Vicuna, an open-sourced LLM to interpret the oracle item. However, we find that our available GPU resources currently can not afford the finetuning.  We will keep investigating this promissing direction. Moreover, in the revised version, we will discuss more about how to fetch the recommended item(s) from the oracle item, and what LLMs can do in the process. Hopefully, it can inspire more research in this direction! Thanks again for this insightful comment!

---

> > ### Author Response · Authors · 2023-08-15
> > **Supplement to Point 6（the additional baseline [21]）**
> >
> > As a supplement to Point 6, we have implemented model [21] based on the paper's details, and the result is shown as follows:
> >
> >  ||YooChoose | YooChoose | KuaiRec | KuaiRec | Zhihu | Zhihu
> > :--: |:--: | :--: | :--: | :--: | :--: | :--:
> >  ||HR@20(%) | NDCG@20(%) | HR@20(%) | NDCG@20(%) | HR@20(%) | NDCG@20(%)
> > DSR [21] | 4.12$\pm$0.06|1.81$\pm$0.05|3.79$\pm$0.03|1.64$\pm$0.03|1.80$\pm$0.04|0.65$\pm$0.02
> >
> > Note that the results of models [19], [20], and [36] are provided in Table 4 of the PDF.
> >
> > We are anticipating a deeper discussion with you!

---

> > > ### Comment · Reviewer_b6yV · 2023-08-15
> > >
> > > Thank you, authors, for the detailed response to my questions and suggestions. Hope some of the discussions would be useful to the paper and will be integrated to it in the final version.
> > >
> > > Given the potential impact of the new recommendation framework proposed by the work, I'll continue voting for accepting the paper.

---

> > > > ### Author Response · Authors · 2023-08-16
> > > >
> > > > Dear Reviewer b6yV，
> > > >
> > > > Thank you for your time and effort in reviewing our paper and providing valuable feedback. We appreciate your insights, and we will try our best to integrate the discussion into our revision.
> > > >
> > > > We sincerely hope that our response has properly addressed your concerns. If so, we would deeply appreciate it if you could re-evaluate our work and raise your score slightly. If you have further concerns, we will continue actively responding to your comments and/or questions. Thank you again for your attention.
> > > >
> > > > Best,
> > > >
> > > > Authors

---

### Official Review · Reviewer_VUDN · 2023-07-06

**Soundness:** 3 good
**Presentation:** 3 good
**Contribution:** 3 good
**Rating:** 6
**Confidence:** 3

**Summary:**

This paper describes a new learning-to-generate paradigm for sequential recommendation based on diffusion models. The authors discuss the limitations of previous approaches in sequential recommendation, where a recommender model learns to classify user preferences based on positive and negative item samples. The paper highlights two inherent limitations of this approach: (1) it differs from human behaviour, which involves knowing an ideal item and selecting potential matches, and (2) the classification is limited to the candidate pool, which may contain noisy or easily distinguishable negative samples, diluting the preference signals. This paper proposes Diff4Rec, a new learning-to-generate paradigm based on guided diffusion, which is able to circumvent the limitations of existing approaches. Empirical results on benchmark datasets show the effectiveness of Diff4Rec.

**Strengths:**

- The authors analyzed the potential limitation of using negative sampling when training classification based recommendation models, and proposed a generation based framework using diffusion models, which is technically sound.
- Diff4Rec is able to outperform a number of state-of-the-art baseline alternatives.
- Code is available, which helps reproduce the results.

**Weaknesses:**

- The authors analyzed the potential limitations of using negative sampling when training classification based recommendation models. In practice, however, a lot of models can be trained without using negative sampling at all. They directly use a softmax over all possible items to obtain the probability of each item being the next item. These softmax based approaches, in my opinion, are closely related to the spirit of Diff4Rec, i.e., by not relying on negative sampling and treat all items other than the next item as a whole. Thus the authors need to explain why Diff4Rec is superior to these approaches.
- Diff4Rec can be very slow in both training and inference stages. In addition, the authors should analyze the time / computational complexity of Diff4Rec and present them in the paper.
- The number of compared approaches is relatively small.

**Questions:**

- In Diff4Rec, the max number of interactions is set to 10. Why not choosing a larger number of historical interactions? Is it due to Diff4Rec running too slow, or is it because of Diff4Rec not performing well when the sequential length is large?
- The abstract in the OpenReview console is not consistent with the submitted paper. In OpenReview, the model is termed SRDiff, while in the paper it is termed Diff4Rec.

**Limitations:**

The authors have done a good job discussing the limitations of Diff4Rec. Diff4Rec has a very high computational cost, which limits its applicability. When considering using Diff4Rec in application, it is important to analyze the tradeoff between computational need and recommendation performance. However, Diff4Rec offers a new view of reshaping sequential recommendation as an item generation task, which has the potential to inspire more research in this direction. This outweighs the existing limitations of Diff4Rec.

---

> ### Author Rebuttal · Authors · 2023-08-09
>
> > 1. *Why Diff4Rec is superior to softmax-based approaches.*
>
>    We appreciate your comments, but respectfully emphasize the fundamental distinction between Diff4Rec and softmax-based approaches. Conceptually, Diff4Rec is a learning-to-generate paradigm that casts off negative sampling, while softmax-based approaches are in the learning-to-classify paradigm that contrasts the positive items against the explicit or implicit negatives.
>
>    We focus on the softmax loss with implicit negatives, which may not conduct the negative sampling explicitly, but implicitly treat all items (excluding the next positive item) as negatives [a]. Specifically, the softmax loss can be derivated as follows (cf. Equation 14 in [a]):
>
>    $L_{softmax} = -\sum\limits_{(c, i) \in S} \ln \frac{\exp(\hat{y}(i|c))}{\sum\limits\_{j \in I}\exp(\hat{y}(j|c))} = -\sum\limits_{(c, i) \in S}\left[ \hat{y}(i|c) - \ln \sum\limits_{j \in I}\exp(\hat{y}(j|c))\right],$
>
>    where $S$ is the observed interactions. Clearly, it treats the observed interactions as positives, while implicitly designating unobserved interactions as negatives. Then, it optimizes their margin appropriately.
>
>    Furthermore, work [b] interprets softmax loss from the perspective of contrastive learning:
>
>    $L_{SSM} = -\sum\limits_{(u, i) \in D} \log \frac{\exp(f(u, i))}{\exp(f(u, i)) + \sum\limits_{j \in N}\exp(f(u, j))},$
>
>    which aligns with softmax loss when $N$ is the set of all items apart from the next positive item.
>
>    Concurrently, softmax loss can be interpreted by InfoNCE, a well-known objective in contrastive learning, by setting the temperature $\tau = 1$:
>
>    $L_{InfoNCE} = -\sum\limits_{(u, i) \in D} \log \frac{\exp(f(u, i) / \tau)}{\exp(f(u, i) / \tau) + \sum\limits_{j \in N^-}\exp(f(u, j) / \tau)},$
>
>    and InfoNCE is acknowledged to discriminate between positive sample (u, i) and negative ones (u, j).
>    The training of Diff4Rec, as shown in Algorithm 1, involves no implicitly usage of negative samples, compared to softmax.  The comparison can be summarized as:
>
>    method|observed interactions|unobserved interactions|treatment
>    :--:|:--:|:--:|:--:
>    softmax |positive|negative (explictly)| contrastive learning
>    Diff4Rec|positive|not use|recover observed interactions by diffusion
>
>    [a] Rendle, S. (2021). Item recommendation from implicit feedback. In Recommender Systems Handbook (pp. 143-171). New York, NY: Springer US.
>
>    [b] Wu, J., Wang, X., Gao, X., Chen, J., Fu, H., Qiu, T., & He, X. (2022). On the effectiveness of sampled softmax loss for item recommendation. arXiv preprint arXiv:2201.02327.
>
> ---
>
> > 2. *The authors should analyze the time / computational complexity of Diff4Rec and present them in the paper.*
>
>    Thank you for the insightful comment. Indeed, the diffusion model's inference phase can be time-consuming—an intrinsic limitation of diffusion. While Diff4Rec reframes sequential recommendation as a learning-to-generate task using diffusion, it is not immune to this limitation. However, we're optimistic that as diffusion model research advances, emerging efficient inference algorithms will mitigate this slow-inference concern. Thus, Diff4Rec stands to benefit from these advancements in diffusion.
>
>    We recognize the importance of analyzing Diff4Rec's computational complexity. Consequently, we provide Table 5 in the PDF, detailing the time taken at each stage. We can observe that the training efficiency of Diff4Rec and SASRec is comparable, since diffusion would sample a step for training as described in Line 5 of Alg. 1. The inference stage of Diff4Rec is more time-consuming than SASRec. Moreover, we show the trade-off between performance and time cost of inference *wrt* total diffusion steps in Figure 8.
>    We'll definitely incorporate these computational costs in the revision. We're grateful for your insightful remarks.
>
> ---
>
> > 3. *Why not choose a larger number of historical interactions than 10?*
>
>    Thank you for raising this point. Our decision to set the maximum number of interactions at 10 was not due to performance or efficiency constraints of Diff4Rec. We thought 10 is an appropriate number. Indeed, Diff4Rec's ability to handle longer sequences is not compromised. In Diff4Rec, historical interactions are initially encoded using a transformer encoder to obtain a 1-D representation, which then guides the diffusion process. As such, the length of retained historical interactions has a limited impact on learning efficiency. To provide further clarity, we conducted experiments with larger sequential lengths (20 and 30). The results are in Table 6&7 in the PDF, which shows similar trends with the results in Table 1. Meanwhile, the training and inference time show little difference with Table 5.
>
>    We hope the results address your concerns.
>
> ---
>
> > 4. *The number of compared approaches is relatively small.*
>
>    Thank you for the suggestion. We've incorporated three additional baseline models (i.e., (DiffRec [36], DiffuRec [19], and CDCR
> ec [20]) in Table 4 of the PDF, inclusive of recent work applies diffusion to recommendation. Note that these models still operate under the learning-to-classify paradigm. As described in Lines 101-110, they necessitate the use of negative samples.
>
> ---
>
> > 5. *The abstract in the OpenReview console is not consistent with the submitted paper.*
>
>    Thank you for bringing this issue to our attention. We apologize and will perform careful proofreading in the revision.
>
> ---
>
> > 6. *It is important to analyze the tradeoff between computational need and recommendation performance.*
>
>    Thank you for your feedback. As mentioned in Point 3, the balance between computational cost and applicability is a fundamental challenge of diffusion models. Following your suggestion, we've presented experimental results and analysis. Moreover, your comments underscore our belief that, despite its limitations, Diff4Rec has the potential to spur further research in this domain.

---

> ### Author Response · Authors · 2023-08-18
> **We are looking forward to your further comments.**
>
> Dear Reviewer VUDN,
>
> Thank you again for your insightful feedback on our submission, particularly your suggestions to 1) **explain why Diff4Rec is superior to softmax-based approaches**, 2) **discuss the time / computational complexity of Diff4Rec in the paper**, 3) **verify our method with a larger number of historical interactions than 10**, and 4) **incorporate more baseline approaches**. These valuable suggestions better strengthen the quality of our work. The deadline of the discussion stage is approaching, and we are looking forward to your further comments.
>
> We sincerely hope that these improvements will be taken into consideration. If we have properly addressed your concerns, we would deeply appreciate it if you could kindly re-evaluate our paper. If you have further concerns, please let us know and we remain open and would be more than happy to actively discuss them with you.
>
> Best,
>
> Authors

---

> > ### Comment · Reviewer_VUDN · 2023-08-18
> >
> > I would like to thank the authors for the detailed explanation. After reading other reviewers' comments and the authors' rebuttal, I will keep my original score.
> >
> > Firstly, softmax-based approaches should also be considered as generative methods since they directly models the probability of a target item being the next item. In that way, the distinction between softmax and diffusion is really marginal.
> >
> > The authors also argued that softmax-based approaches explicitly used negative samples, while diff4rec does not. In fact, both softmax and diffusion implicitly use negative sampling. If a diffusion model does not implicitly use negative sampling, then the model will simply collapse.
> >
> > Another potential problem of the evaluation protocol is that, the authors said "For all baselines, we conduct negative sampling from the uniform distribution at the ratio of 1: 1, which is not conducted in Diff4Rec." This is not really a good practice for training sequential recommendation models. A negative size of 1 is simply too small for the model to perform well. In my experience, for example, on the YooChoose dataset, a negative sample size of 49-99 would achieve much better result. Therefore the results reported in the paper might not be a fair comparison against baseline methods.

---

> > > ### Author Response · Authors · 2023-08-18
> > > **Responce to your further concerns.**
> > >
> > > Dear Reviewer VUDN,
> > >
> > > Thank you very much for your valuable feedback. We sincerely hope that our following response could properly address your concerns.
> > >
> > > 1. >*Softmax-based approaches should also be considered as generative methods since they directly model the probability of a target item being the next item. In that way, the distinction between softmax and diffusion is really marginal.*
> > >
> > > We do agree that softmax directly models the probability of a target item being the next item. Meanwhile, we would like to kindly emphasize that **softmax and generative models are different due to their adherence to distinct paradigms (discriminative and generative respectively)**.
> > >
> > > Specifically, generative models (GANs, VAEs and diffusion models, e.t.c) directly model the underlying data generation distribution by learning the map from Gaussian distribution and the underlying distribution. **In contrast, softmax is commonly employed in discriminative models, since it models the probability distribution over a set of discrete classes, with limited presence in the literature of generative models**. Besides, softmax is limited to the discrete candidate set, whereas diffusion can generate samples beyond the candidate. Therefore, we would respectfully emphasize that softmax and diffusion models are quite distinctive.
> > >
> > > 2. >*The authors argued that softmax-based approaches explicitly use negative samples, while diff4rec does not. In fact, both softmax and diffusion implicitly use negative sampling. If a diffusion model does not implicitly use negative sampling, the model will simply collapse.*
> > >
> > > We agree that softmax implicitly uses negative sampling (we feel sorry about the potential confusion caused by that it should be 'implicitly' instead of 'explicitly' in how softmax uses unobserved interactions in the table of Point 1 in the rebuttal). Meanwhile, we would kindly emphasize that **negative sampling is not used in Diff4Rec, either explicitly or implicitly**. The primary objective of Diff4Rec is:
> > >
> > > $L_{t-1} = \mathbb{E}_{e_n\^0, \epsilon}\left[\frac{\bar\alpha\_{t-1}}{2\tilde{\beta}\_t} ||e\_n^0 - f\_\theta(\sqrt{\bar\alpha\_t} e_n^{0} + \sqrt{1-\bar\alpha\_t}\epsilon, c\_{n-1}, t)||^2\right] + C,$
> > >
> > > where $e_n^0$ is sampled from **only observed interactions**. We have also provided the code and training algorithm, and we hope these materials could help address your concern.
> > >
> > > **Even if negative sampling is not used (implicitly or explicitly) in Diff4Rec, it would not collapse, which is one of the contributions of this work**. The reason can be attributed to the fundamental paradigm shift: the paradigm of Diff4Rec is learning-to-generate through diffusion, instead of traditional learning-to-classify. Specifically, Diff4Rec distinctly models the underlying generation distribution of observed interactions with the power of diffusion, which does not require negative samples. However, previous sequential recommenders adhere to the learning-to-classify paradigm and distinguish positive and negative samples, which strictly requires negative sampling.
> > >
> > > **We do understand that recommendation has long been recognized as a discriminative task requiring negative sampling. Yet our work does show that sequential recommendation can be reshaped as a generative task with diffusion, and discard negative sampling.**
> > > We sincerely hope that the initial exploration of Diff4Rec could enable recommendation task to embrace the benefits offered by the rapid development of diffusion models.
> > >
> > > 3. >*A negative size of 1 is too small for the model to perform well. In my experience, on the YooChoose dataset, a negative sample size of 49-99 would achieve much better result. Therefore the results reported in the paper might not be a fair comparison against baseline methods.*
> > >
> > > We recognize the significance of a fair experimental setting, particularly the number of negative samples. In the literature of recommendation, **a fair experimental setting regarding the number of negatives could be that the number of negatives keeps the same across all models, addressing any potential bias arising from varying numbers of negatives.** Given that Diff4Rec discards negative samples, and the number of negatives of Diff4Rec can be seen as 0, we believe it is justifiable for other baselines to set their number of negatives as 1.
> > >
> > > Meanwhile, we acknowledge the concern that more negative samples in classification-based baselines can improve performance, and we conduct experiments to search the number of negatives in [50, 60, 70, 80, 90, 100] following your advice. The results are as follows:
> > >
> > > ||YooChoose |-|KuaiRec|-|Zhihu|-
> > > :--:|:--:|:--:|:--:|:--:|:--:|:--:
> > > ||HR@20(%)|NDCG@20(%)|HR@20(%)|NDCG@20(%)|HR@20(%)|NDCG@20(%)
> > > SASRec|4.18|1.79|4.12|1.96|1.87|0.69
> > > CL4SRec|4.64|1.93|4.31|2.11|2.10|0.75
> > > Diff4Rec|4.78|2.23|5.26|4.11|2.26|0.79
> > >
> > > Note that CL4SRec constructs many negative samples with augmentation, therefore more negatives result in limited improvement.
> > >
> > > Best,
> > >
> > > Authors

---

### Official Review · Reviewer_RbjC · 2023-07-10

**Soundness:** 2 fair
**Presentation:** 3 good
**Contribution:** 3 good
**Rating:** 5
**Confidence:** 3

**Summary:**

The paper presents a new approach Diff4Rec, a guided diffusion model for sequential recommendations. The paper could be divided in three main parts:
1. Problem formulation and method: It describes the sequential recommendation as oracle item generation and then explains how diffusion is applied.
2. Experiments: Describes the datasets used and various experiments done. Moreover, they have provided code also.
3. Results: Presents the results from their experiment. The results are much superior compared to the previous methods.

**Strengths:**

The paper presents a new idea in the learning-to-generate paradigm, which aimed to used diffusion process. They have supported the results with exhaustive set of experiments and validated the results on multiple datasets.

**Weaknesses:**

1. The paper doesn't show the impact of negatives on the recommendations
2. Would it be possible to present the A/B test results. I suspect the recommendations would over hinge on the users history while recommending for a longer period of time..

**Questions:**

1. Noob q: As per my understanding, during the inference time we will get embedding for nth item as a oracle item. How do we recover what item to recommend based on the predicted embedding?
2. Do we have any results from an A/b test?

**Limitations:**

A/B testing on real system would have helped to validate the proposed idea.

---

> ### Author Rebuttal · Authors · 2023-08-09
>
> > 1. *The paper doesn't show the impact of negatives on the recommendations.*
>
>    Thank you for raising this point. We feel a little confused about the "impact of negatives", and try to analyze it from two distinct perspectives:
>
>    - **Negative Societal Impacts**: we would like to discuss the negative social impact of recommendation. Firstly, one major concern about recommender system is the potential of privacy disclosure and information leakage, and it is not a risk in our work since the datasets are all anonymized by the provider. Secondly, recommender system may bring issues such as Information cocoons and echo chambers, which are also significant research topics beyond the scope of our work.
>
>    - **Impact of Negative Sampling**: we would like to discuss more about negative sampling in recommender system. As described in Line 44-52 and Figure 1, the learn-to-classify based recommenders are demanding of negative samples to discriminate between positive samples and negative ones when learning the decision boundary.  Without negative sampling, the item embedding of learn-to-classify based recommenders would be pathologically distributed as shown in Figure 3 of the main text and Figure 6 of the Supplementary Material. Diff4Rec, by contrast, explores the underlying distribution of observed interactions with diffusion model, and does not require negative samples in the learning process.
>
>    We eagerly anticipate a deeper discussion on these matters in the subsequent stages!
>
> ---
>
> > 2. *How do we recover what item to recommend based on the predicted embedding?*
>
>    Thank you for bringing up this point! As mentioned in Lines 253-256, once we obtain the embedding of the oracle item, our next step is to identify the top K-nearest items from the candidate set for top-K recommendation, based on their similarity (i.e., inner product between the embeddings of the oracle and candidate). We acknowledge the significance of this step in achieving the recommendation task, and to provide a comprehensive understanding, we will detail this step in Method of the revision. We're grateful for your astute feedback, which has undeniably enriched our presentation.
>
> ---
>
> >  3. *Would it be possible to present the A/B test results? I suspect the recommendations would over hinge on the users history while recommending for a longer period of time.*
>
>    Thank you for bringing this point to us. We recognize the potential of A/B testing to validate the efficacy of recommendation models, though
> our current access does not extend to online A/B testing platforms.
>
>    Moreover, we notice that the main concern comes from that the recommendation would over hinge on the user history while recommending for a longer period of time. To address this, we further simulate the A/B testing with the assistance of ChatGPT, inspired by its remarkable generalization and simulation ability, since it encodes a wide range of human behavior from their training data  [a]. Such ChatGPT-simulated A/B testing is less prone to the user history, as compared to the conventional evaluation on the fixed test data. We list the steps:
>
>    - Data Split: We sample 100 real users from the MovieLens-100k dataset for evaluation, reserving the rest as the training data;
>    - Recommender Training: We train Diff4Rec and CL4SRec (the best baseline model under top-K evaluation) on the training data;
>    - User Simulation: For each user being evaluated, we convert his/her history to a textual prompt to profile the user, and feed it into ChatGPT to simulate the user. Then, three movie lists derived from Diff4Rec, CL4SRec, and Random are presented to ChatGPT to select which list he/she would prefer. We conduct the simulation five times per user with different ChatGPT accounts, and the list with the most votes will have its *Success Score* plus one.
>    - Simulated A/B Testing: We compare the *Success Score* among Diff4Rec, CL4SRec, and Random, based on the selection of ChatGPT-simulated users. The results are shown as follows (Note that two lists may have even votes (3 out of 100), and we denote this inside the bracket):
>    Random | CL4SRec | Diff4Rec
>    :--:|:--:|:--:
>    12|33(3)|52(3)
>
>    Clearly, Under the evaluation of ChatGPT-simulated users, Diff4Rec performs better than CL4SRec. Considering that this evaluation is not based on the pre-collected test data,  Diff4Rec is less likely to over hinge on users history.
>
>    [a]Park, J. S., O'Brien, J. C., Cai, C. J., Morris, M. R., Liang, P., & Bernstein, M. S. (2023). Generative agents: Interactive simulacra of human behavior. arXiv preprint arXiv:2304.03442.
>
> We're grateful for your insightful suggestions, which have enriched our evaluations and inspired us for future directions.

---

> ### Author Response · Authors · 2023-08-18
> **We are looking forward to your further comments.**
>
> Dear Reviewer RbjC,
>
> Thank you again for your valuable feedback on our submission, particularly your suggestions to 1) **highlight the inference of recommendation results**, and 2) **demonstrate that the recommendations would not over-hinge on the users' history with A/B test**. We also try our best to address your concern about **the impact of negatives on the recommendations from the perspectives of negative societal impacts and the impact of negative sampling**. These insightful suggestions better strengthen the quality of our paper. The deadline of the discussion stage is approaching, and we are looking forward to your further feedback.
>
> We sincerely hope that these improvements will be taken into consideration. If we have properly addressed your concerns, we would be grateful if you could kindly re-evaluate our paper. If you have additional concerns, please let us know and we remain open and would be more than happy to actively discuss them with you.
>
> Best,
>
> Authors

---

### Author Rebuttal · Authors · 2023-08-10

We thank the reviewers for their insightful and positive feedback! We are encouraged that they find Diff4Rec introduces a transformative methodology in sequential recommendation (Reviewer $\color{Blue}\text{b6yV}$), achieves much superior performance compared to state-of-the-art baseline alternatives (Reviewer $\color{Goldenrod}\text{RbjC}$ and Reviewer $\color{Red}\text{VUDN}$), potentially broadening the capabilities of recommendation systems (Reviewer $\color{Blue}\text{b6yV}$),  and has the potential to inspire more research in this direction (Reviewer $\color{Red}\text{VUDN}$).

We would also like to express our gratitude to the reviewers for highlighting that Diff4Rec presents an innovative paradigm shift in sequential recommendation systems (Reviewer $\color{Blue}\text{b6yV}$), effectively mirrors human behavior (Reviewer $\color{Blue}\text{b6yV}$), and is technically sound (Reviewer $\color{Red}\text{VUDN}$).

After carefully analyzing the reviewers' comments, we are more than glad to find these comments very insightful. We respond to the comments point by point in the rebuttal and provide a PDF containing the comparison with more baseline models and the demonstration of the efficiency of Diff4Rec. We hope that our response can address the concerns raised by the reviewers.

Once again, we sincerely thank the reviewers for their valuable feedback and insightful suggestions, which undoubtedly contribute to enhancing the quality of our work.

We eagerly anticipate the ensuing discussions in the next phase!

---

### Author Response · Authors · 2023-08-21
**Thank you for your time and effort.**

Dear AC and Reviewers,

We would like to express our sincere gratitude for your dedicated time and efforts in evaluating our submission.

In essence, **Diff4Rec reshapes sequential recommendation as a learning-to-generate task with diffusion models, in contrast to the conventional perception of recommendation as a learning-to-classify (or discriminative) task**. Therefore, Diff4Rec has the potential to explore the underlying distribution of item space, enabling the generation of items beyond the confines of predefined candidates. This sets Diff4Rec apart from classification-based recommenders, which are constrained by discrete candidate set.

Having received so many valuable suggestions in the rebuttal and discussion phases, we are committed to  incorporating the reviewers' suggestions into our manuscript. These insightful suggestions would definitely strengthen the quality of our paper! We are particularly appreciative of the reviewers' recognition that Diff4Rec brings the potential impact of the new recommendation framework , and the potential to inspire more research in this direction.

Once again, we sincerely express our gratitude to you, our AC and reviewers, for guiding us through this insightful review process!

Appreciatively,

Authors

---

### Decision · Program_Chairs · 2023-09-21

**Decision:**

Accept (poster)

**Comment:**

This paper presents Diff4Rec: a diffusion based model for sequential recommendation. The basic idea is straightforward: in sequential recommendation setting, we are ultimately interested in estimating the conditional density p(next item | historical items), which has mostly been modeled under a classification task. Instead, Diff4Rec considers a generative process where we estimate this density in a continuous embedding space with a diffusion model and sample from it, which will generate an embedding that doesn't correspond to any actual item but we can make recommendations based on the proximity in the embedding space. The paper received borderline positive scores. All the reviewers agree that the learning-to-generate perspective of Diff4Rec is interesting and the empirical results seem convincing. However, the reviewers also shared some concerns around the practicality, given the expensive generation process of the diffusion models, as well as the argument of "no negative sampling" for Diff4Rec. I read the paper myself and I can see how Diff4Rec can get away with not using negative examples, even though I wouldn't necessarily consider this paradigm new, since this argument is not that dissimilar to the "generative vs discriminative ML" arguments from back in the days, which I hope the authors can acknowledge.

Overall considering this is, as far as I know, the first work on using a diffusion model as a generative model for sequential recommendation (as opposed to a discriminative model) with reasonably good empirical support, even though the technical contribution is relatively limited, I am OK with accepting this paper (albeit very weakly). However, I want the authors to tune down some of the claims unless more justification can be provided. For example, I especially find the authors' argument about an "oracle item" not very scientific: quoting the abstract "...(1) it may differ from human behavior that user usually imagine an oracle item in mind and selects potential items matching the oracle; and (2) the classification is limited in the candidate pool with noisy or easy supervision from negative samples, which dilutes the preference signals towards the oracle item." Is there any real evidence backing these (especially the human behavior part) up? Or is this just to match the generative process of Diff4Rec? Similarly, the argument of "learning-to-generate" vs "learning-to-classify" is not something groundbreakingly new either -- as I mentioned before, this dates back to before deep learning as "generative vs discriminative models".

The reviewers have also brought up some comments/suggestions which I hope the authors can incorporate into the final version of the paper.